# Varicella-Zoster Virus Infection and Varicella-Zoster Virus Vaccine-Related Ocular Complications

**DOI:** 10.3390/vaccines13080782

**Published:** 2025-07-23

**Authors:** Jing Yu, Huihui Li, Yuying Ji, Hailan Liao

**Affiliations:** 1Ophthalmology Department of the Second Affiliated Hospital, School of Medicine, The Chinese University of Hong Kong, Shenzhen & Longgang District People’s Hospital of Shenzhen, Shenzhen 518172, China; yuj034@126.com (J.Y.); lizhenyu925@126.com (H.L.); liaohailan888@126.com (H.L.); 2State Key Laboratory of Ophthalmology, Zhongshan Ophthalmic Center, Sun Yat-sen University, Guangdong Provincial Key Laboratory of Ophthalmology and Visual Science, Guangzhou 510060, China

**Keywords:** varicella-zoster virus, ocular complications, vaccine

## Abstract

The varicella-zoster virus is a human herpesvirus that causes varicella as the primary infection and HZ as the reactivation of a latent infection. Ten to twenty percent of cases of herpes zoster ophthalmicus (HZO) involve the ophthalmic branch of the fifth cranial nerve. Any area of the eye may be affected by the condition. HZ has a lifetime risk of more than 30%. Complications from herpes zoster can significantly lower quality of life. The goal of HZ vaccinations is to stop HZ activation and PHN formation. Despite the uncommon possibility of side effects such as eye problems, the majority of vaccines on the market now are safe. The purpose of this review is to discuss VZV infection and analyze and summarize the ocular complications following VZV vaccination.

## 1. Introduction

Varicella-zoster virus (VZV) is a neurotropic human herpesvirus and a double-stranded DNA virus that can cause two distinct diseases: varicella (chickenpox) and herpes zoster (shingles), a vesicular dermatomal rash brought on by the latent virus reactivating. Varicella zoster virus-specific T cell-mediated immunity is largely in charge of regulating viral reactivation and causing herpes zoster. Months to years after the initial infection resolves, VZV establishes latency in peripheral ganglia neurons. Endogenous (subclinical) reactivation and exogenous re-exposure to VZV periodically increase immunity, and VZV-specific memory T cells, which have a mixed central and effector nature, are crucial for preserving VZV latency [1]. A latent virus can nevertheless replicate and produce recurring clinical illness [2]. Declining T cell immunity, not humoral immunity, is primarily responsible for the rise in VZV reactivation seen with aging [3]. Regardless of varicella incidence, herpes zoster is a rare, year-round condition with no seasonal predominance. There is no proof that HZ is caused by an external VZV infection, meaning that it can be contracted by coming into contact with an HZ or varicella carrier. Instead, variables affecting the virus–host relationship—most notably the host’s VZV-CMI, which preserves VZV latency—determine the prevalence of HZ. Even in nations where varicella vaccination has all but eradicated the disease, a number of epidemiologic studies have demonstrated that the age-specific incidence of HZ has risen in the previous seven decades [4,5,6,7].

HZ has a lifetime risk of more than 30%. The incidence rate of HZ in people aged 50 and older varies from 5.23 to 10.9/1000 person-years [8] and from 3 to 5/1000 person-years globally [9]. As people age, the frequency and severity of HZ rise. Nearly half of older adults with HZ experience problems [9], and 50% of those who live to age 85 will acquire HZ [9,10,11]. Herpes zoster and its aftereffects can cause irreparable loss of independence and significantly lower quality of life [12]. Postherpetic neuralgia (PHN), the most common disabling consequence of HZ, is characterized by neuropathic pain and dysesthesia that persists for weeks, months, or even years after the rash has healed [13,14,15,16]. The frequency of PHN rises significantly with age [13,17]. The development of vaccinations to protect older adults from HZ is prompted by the rising incidence of HZ and its crippling consequences as people age. The FDA has approved a number of vaccines to prevent VZV. A live attenuated virus called Varivax is used to prevent varicella in children [18]. In 2006, the FDA authorized Zostavax, a live attenuated Oka strain virus, for the secondary prevention of zoster. However, because of the risk of zoster reactivation in immunocompromised persons and its gradual decline in immunogenicity, its usage has been restricted [19]. The FDA authorized Shingrix, a recombinant subunit zoster vaccine, in 2017, for the prevention of zoster in immunocompetent people 50 years of age and older. Clinicians should be aware of the possibility of post-vaccination VZV infection or reactivation, even if it seems to be uncommon. A pertinent overview of VZV infection and ocular problems linked to VZV vaccination is hence the goal of this work.

## 2. VZV Infection and Common Related Ocular Complications

### 2.1. VZV Infection

Primary VZV infection results in varicella, which is typified by a widespread itchy rash that quickly develops from macules to papules to vesicular lesions and finally crusts. VZV creates a permanent delay in the sensory and autonomic ganglia during the initial infection [20,21]. HZ, a localized illness of the skin, nerve, and sensory ganglia, is caused by VZV replication after reactivation from latency. Herpes zoster typically only affects the dermatome innervated by a single dorsal root or cranial nerve ganglion, resulting in unilateral radicular discomfort and a vesicular rash [22]. The clustering of the lesions indicates intraneural transmission to the skin. Regardless of varicella incidence, herpes zoster is a rare, year-round condition with no seasonal predominance.

### 2.2. Common Ocular Complications Associated with VZV Infection

Ocular problems can occasionally be linked to primary VZV infection in children. In 12–25% of cases, a moderate acute anterior uveitis may develop, resulting in discomfort, perilimbal injection, swelling of the lids, irritation, photophobia, and reduced visual acuity [23]. Furthermore, there have been documented instances of acute retinal necrosis (ARN) complicating a primary VZV infection [24].

Herpes zoster ophthalmicus (HZO) is the involvement of the ophthalmic branch of the trigeminal nerve, and it represents 10–20% of HZ cases [25]. The disease can impact any part of the eye, including the conjunctiva, cornea, trabecular meshwork, and uvea, which are frequently the sites of immune-mediated injury as well as direct viral invasion. Due to the paralysis of the orbicular muscle, the virus can cause lagophthalmos, which is the inability to close the eyelids completely, as well as hyperemia, edema, skin rashes, ptosis, and impaired palpebral motility [26,27,28]. Conjunctivitis was the most prevalent manifestation at the conjunctival level. Corneal surface epithelial keratitis [26,28], pseudodendrites [27,28,29], anterior stromal infiltrates [26,28], late corneal mucous plaque keratitis (MPK) [30], disciform keratitis [26,28], endothelitis with possible endothelial cell loss [31], neurotrophic keratitis due to sensory nerve involvement with corneal hypoesthesia or anesthesia [29,32,33], or exposure keratitis, if linked to an eyelid defect [26], are among the ocular structures most commonly affected by HZO. Scleritis and episcleritis [27,34,35], cataracts [29], anterior uveitis [27,28,32,36], and iris degeneration with sectoral atrophy [27,28] are further signs of HZO. High intraocular pressure during HZO is frequently observed and may be associated with subsequent glaucoma [27,29,37] and trabeculitis [28]. An uncommon side effect of HZO is optic neuritis [32]. VZV is also the most common cause of ARN, as was previously indicated.

According to reports, 6.6–10% of patients experience a reduction in visual acuity as a result of HZO due to possible ocular problems [27,34,38].

## 3. Varicella-Zoster Virus Vaccines and Associated Ocular Complications

### 3.1. Varicella-Zoster Virus Vaccines

The FDA has approved a number of vaccines to prevent VZV. A live attenuated virus called Varivax (Merck, Whitehouse Station, NJ, USA) is used to prevent varicella in children [18]. Since the varicella immunization program was put into place in 1995, there has been a notable decline in both the incidence of varicella and hospitalizations attributable to the disease [39]. Rarely, keratitis or anterior uveitis have been reported in the days after vaccination. It is unclear to determine whether the ocular issues in the majority of these cases were caused by the live attenuated virus that was injected or by the wild-type virus’s pre-existing latency, notwithstanding the temporal link [40,41].

For the secondary prevention of zoster, Zostavax (Merck, Whitehouse Station, New Jersey) is a live attenuated Oka strain virus that was approved by FDA in 2006. The live attenuated virus is the same in both Zostavax and Varivax, but Zostavax has a larger dosage. Zostavax decreased the “burden of illness due to HZ” by 61.1%; decreased the “incidence of clinically significant PHN” by 66.5%; and decreased the “incidence of HZ” by 51.3% [42]. Furthermore, Zostavax markedly decreased the adverse impact of HZ on quality of life and capacity to perform activities of daily living. Additionally, Zostavax significantly reduced the negative effects of HZ on quality of life and ability to carry out daily tasks [43].

However, after receiving the Zostavax vaccine, some individuals with a history of HZO may experience ocular, dermatological, or widespread recurrence, according to some studies [27,35,44]. However, because of the risk of zoster reactivation in immunocompromised persons and its gradual decline in immunogenicity, its usage has been restricted. In many nations today, the recombinant zoster vaccine Shingrix has largely replaced Zostavax. Shingrix is a subunit vaccination that includes the AS01B adjuvant system together with the VZV glycoprotein E antigen. For individuals aged 50 and over, Shingrix is a new, highly effective, and well-tolerated vaccine alternative that reduces the incidence of HZ (more than 90% decrease in risk of HZ) and postherpetic neuralgia. According to US and Canadian guidelines, it is recommended above a live attenuated HZ vaccination in immunocompetent persons and is not contraindicated in immunocompromised subjects [45]. One rare but dangerous possible ocular side effect of recombinant zoster immunization is uveitis recurrence. There have been numerous reports of uveitis reactivation after Shingrix. It is quite uncommon for HZO to develop after Shingrix.

### 3.2. Ocular Adverse Events Following VZV Vaccination

Although they have many advantages, vaccines can potentially have negative effects. Among these problems, uveitis and ocular inflammation are rather common [46]. We have summarized and compared all the cases in Table 1 and Table 2.

According to previously published reports, 20 cases comprising 24 patients with VZV-associated ocular complications have been described [47,48,49,50,51,52,53,54,55,56,57,58,59,60,61,62,63,64,65,66]. The average age of patients after Varivax was 17.2 years (range, 5–42 years), and the average age of patients after Herpes Zoster Vaccines was 69.1 years (range, 50–89 years), with a male-to-female ratio of 7:5; 45.8% patients presented systemic symptoms, and 79.1% of patients experienced varying degrees of decline in visual acuity, with the onset of symptoms varying from 24 h to 3 years. In total, 79.2% of patients had underlying medical conditions. Corticosteroids and/or antiviral medications were used to treat every case. Except for one patient who died from the primary disease [59], one patient with significant visual decline [62], and one patient who developed secondary neovascularization [63], the remaining patients recovered well.

Only 54.2% of the cases underwent VZV DNA testing, and the results showed that three of these patients tested positive for the Oka strain of the VZV vaccine [52,55], pointing to a possible connection between ARN and vaccination strain infection. And five of these patients had wild-type VZV, suggesting that VZV reactivation rather than vaccination reactivation may have been the cause of the majority of ARN cases. Additionally, the majority of patients had cirrhosis, diabetes, and inflammatory gastrointestinal disorders, which are systemic metabolic conditions that may have contributed to the development of ARN following VZV immunization.

After receiving the VZV vaccine, patients who develop uveitis frequently experience serious systemic illness, indicating that those who have serious underlying illnesses or conditions affecting their immune systems may be more vulnerable to negative reactions to the VZV vaccine. Potential vulnerability to attenuated VZV vaccinations could be the cause of this.

Previous history of HZO is identified by other studies as a risk factor for a potential recurrence after vaccination [50,53,54]. Among our cases, 8 involved recurrence or worsening of pre-existing conditions following vaccination. But according to a study based on the Health-Claim Database [67], patients who received Zostavax did not have a higher risk of anterior segment complications than those who were initially diagnosed with HZ. It was also noted that the reactivation of the virus and HZ after Zostavax vaccination is extremely uncommon. Therefore, despite these possible hazards, the vaccine is regarded as safe [68], problems are uncommon [69], and a history of HZO does not actually preclude vaccination [25,35,50,53,70,71]. According to Shingrix’s post-licensure surveillance, there were few complaints of uveitis and a reporting rate of 0.6/100,000 for inflammatory eye disorders [72]. Potential immune-mediated disorders occurred at a similar rate for both vaccine recipients and controls at all time points, according to the results of two sizable randomized placebo-controlled phase 3 studies of the Shingrix [73]. Similarly, when compared to controls, individuals who already had potential immune-mediated disorders did not show a higher risk of developing a new immune-mediated process or experiencing an exacerbation of their pre-existing condition following immunization.

**Table 1 vaccines-13-00782-t001:** Patients with complications after Varivax.

Study	Diagnosis	Vaccine Name	Vaccine Type	Age	Gender	Signs	Symptoms	Lab Tests	Medical History	Interval Post-Vaccination ^	Treatment	Outcome
Esmaeli-Gutstein et al. [47]	Anterior and intermediate uveitis (left eye)	varicella vaccine	Live attenuated	16	F	generalized vesicular rash on her face and trunk	photophobia, blurred vision, and redness in the left eye	none	none	1 week after vaccination	po acyclovir, topical corticosteroids	resolved completely
Naseri et al. [48]	Herpes zoster virus sclerokeratitis and anterior uveitis (left eye)	varicella vaccine	Live attenuated	9	M	rash in left face	red left eye	wild-type VZV DNA (+)	mild childhood asthma and mild eczema	3 years after vaccination	po acyclovir/topical corticosteroids	faint anterior stromal scar
Fine et al. [49]	Bilateral APMPPE	varicella vaccine	Live attenuated	11	M	severe headaches and tinnitus	blurry vision and photopsias	VZV Ab (+)	none	10 days after vaccination	po acyclovir/oral corticosteroids	partially resolved
Gonzales et al. [52]	Bilateral ARN	varicella vaccine	Live attenuated	20	M	none	red eyes and blurry vision	Oka strain VZV DNA (+)	immunosuppressant for an inflammatory gastroenteropathy	1 month after vaccination	io foscarnet, antiviral drugs, pars plana vitrectomy	undisclosed
Hayat et al. [66]	Bilateral ARN	varicella vaccine	Live attenuated	42	M	generalized vesicular rash, malaise, arthralgia, and body aches	blurred vision	Oka strain VZV DNA (+)	previously undiagnosed human immunodeficiency virus infection	4 weeks after vaccination	io /iv foscarnet, antiviral drugs	visual acuity improved
Andrade et al. [65]	Uveitis	varicella vaccine	Live attenuated	5	M	vesicular cutaneous lesions	hyperemia	VZV DNA (+)	steroid-dependent nephrotic syndrome	14 days after vaccination	iv/po antiviral drugs, steroids, topical corticosteroid	clinically stable

^ Time between last vaccination and initial ocular symptom/sign.

**Table 2 vaccines-13-00782-t002:** Patients with complications after Herpes Zoster Vaccines.

Study	Diagnosis	Vaccine Name	Vaccine Type	Age	Gender	Signs	Symptoms	Lab Tests	Medical History	Interval Post-Vaccination ^	Treatment	Outcome
Khalifa et al. [50]	Exacerbation of Zoster Interstitial Keratitis	ZOSTAVAX	Live attenuated	50	F	none	vision loss, extensive epithelial edema, and diffuse stromal haze involving the lower two-thirds of the left cornea	none	Zoster Interstitial Keratitis	35 days after vaccination	oral valacyclovir, topical corticosteroid	undisclosed
Charkoudian et al. [51]	ARN (left eye)	ZOSTAVAX	Live attenuated	77	F	none	severe vision loss	VZV DNA (+)	diabetes mellitus with secondary end-stage renal disease	6 days after vaccination	po/iv antiviral drugs, vitrectomy	undisclosed
	Bilateral ARN	ZOSTAVAX	Live attenuated	80	M	rash and fever	severe vision loss	VZV DNA (+)	hypertension and renal transplantation	2 months after vaccination	po/iv antiviral drugs, io foscarnet, bilateral vitrectomy	undisclosed
Sham et al. [53]	Exacerbation of anterior uveitis (right eye)	ZOSTAVAX	Live attenuated	86	M	none	vision loss and worsened corneal edema	none	medical history of HZO with anterior uveitis	3 weeks after vaccination	po valacyclovir, topical corticosteroids	return to his baseline condition
Hwang et al. [54]	Reactivation of Herpes Zoster Keratitis	ZOSTAVAX	Live attenuated	63	M	none	redness, pruritus, photophobia, and pain	none	non-Hodgkin lymphoma and right-sided HZO	2 weeks after vaccination	oral valacyclovir, topical corticosteroid	punctate epithelial keratopathy and a mild subepithelial haze
Heath et al. [55]	ARN (left eye)	ZOSTAVAX	Live attenuated	78	F	none	floaters	Oka strain VZV DNA (+)	noteworthy for rheumatoid arthritis, latent autoimmune diabetes of adulthood and osteoporosis	6 weeks after vaccination	po valaciclovir, topical corticosteroids, pars plana vitrectomy	A pigmented scar
Jastrzebski et al. [56]	ARN	ZOSTAVAX	Live attenuated	67	F	none	central corneal staining, corneal perforation	none	recurrent unilateral herpes zoster keratouveitis	2 weeks after vaccination	oral famciclovir/ penetrating keratoplasty	residual pigmented keratic precipitates on the corneal endothelium and punctate epithelial erosions
Ali et al. [57]	ARN and contralateral cutaneous eruption	ZOSTAVAX	Live attenuated	63	M	varicella skin eruption on the right side of his face	insidious deterioration in vision	VZV DNA (+)	none	3 months after vaccination	io foscarnet, po antiviral drugs, po steroids	none
Lehmann et al. [58]	Reactivation of Herpes Zoster Stromal Keratitis	SHINGRIX	Recombinant zoster vaccine	89	M	none	diffuse stromal edema, anterior stromal granular infiltrate, and keratic precipitates	none	herpes-zoster-associated stromal keratitis	3 weeks after the first dose	topical corticosteroid	resolved
Weinlander et al. [59]	ARN (left eye)	ZOSTAVAX	Live attenuated	64	M	none	floaters and cloudy vision	Wild-type VZV DNA (+)	metabolic syndrome and impaired glucose tolerance	16 months after vaccination	po valacyclovir, po/topical corticosteroids	stable region of chorioretinal atrophy
	ARN (left eye)	ZOSTAVAX	Live attenuated	62	M	none	floaters and blurred vision	Wild-type VZV DNA (+)	end-stage liver disease and diabetes mellitus type 2	7 months after vaccination	po valacyclovir, topical corticosteroids	died from complications of his cirrhosis
Heydari-Kamjani et al. [60]	Bilateral uveitis sarcoidosis	SHINGRIX	Recombinant zoster vaccine	53	F	headaches	left eye redness, photophobia, and eye pain	none	mild persistent asthma	4 days after vaccination	topical corticosteroid	none
Chen R.I. et al. [61]	ARN (left eye)	SHINGRIX	Recombinant zoster vaccine	65	F	systemic vesicular rash and hypoxic respiratory failure	worsening floaters and blurred vision	wild-type VZV DNA (+)	immunomodulator for multiple myeloma	6 weeks after vaccination	io foscarnet, iv/po antiviral drugs	free of active retinitis
Menghini et al. [62]	ARN with obliterative angiopathy (left eye)	ZOSTAVAX	Live attenuated	76	M	none	blurry vision, slight pain, and redness	wild-type VZV DNA (+)	insulin-dependent diabetes mellitus, chronic lymphocytic leukemia	2 days after vaccination	io foscarnet, iv/po/iv antiviral drugs, iv/po steroids	left eye visual acuity dropped to perception only
Richards et al. [63]	Recurrent bilateral multifocal choroiditis	SHINGRIX	Recombinant zoster vaccine	57	F	arm swelling at the injection site, chills, malaise, subjective fever, and tinnitus	acute decrease in vision in the right eye (OD) and new metamorphopsia in the left eye	none	immunosuppressant for multifocal choroiditis	24 h after the first dose	po steroids and continued methotrexate	intravitreal bevacizumab for a secondary choroidal neovascular membrane
	Recurrent bilateral anterior and mild intermediate uveitis	SHINGRIX	Recombinant zoster vaccine	69	M	headache	blurred vision	none	idiopathic recurrent bilateral anterior and mild intermediate uveitis	1 month after the second dose	po valacyclovir/topical corticosteroids	none
	Recurrent anterior uveitis (left eye)	SHINGRIX	Recombinant zoster vaccine	70	F	none	mildly decreased vision	none	recurrent unilateral anterior uveitis and corneal neovascularization	2 weeks after the first dose	oral valacyclovir, topical corticosteroid	return to quiescence
R. T. Liu et al. [64]	Reactivation of herpes zoster keratitis	SHINGRIX	Recombinant zoster vaccine	75	F	none	decreased visual, corneal folds	none	HZO keratitis	Two and a half weeks	oral valacyclovir, topical corticosteroid	completely resolved

^ Time between last vaccination and initial ocular symptom/sign.

The underlying mechanism of ocular complications following varicella-zoster vaccination remains unclear. There are different views on different vaccines. There are two possible explanations for the activation of ocular complications after vaccination with live attenuated vaccine: one possibility is that some patients may have immunosuppression, leading to vaccine strain virus infection; the second possibility is that vaccine strain viruses are rarely detected in patients who develop varicella rash after vaccination, and these viruses are more likely to represent the reactivation of varicella [42].

Theoretically, Shingrix cannot cause iatrogenic infection because it is a subunit vaccine rather than an attenuated live virus. It contains the AS01B adjuvant-formulated VZV glycoprotein-E. There could be two possible pathways, as opposed to the direct infection from attenuated but still active live attenuated vaccinations that cause uveitis: (1) Several in vitro investigations, including those involving monophosphorylate lipid A (MPL), have shown that the adjuvant AS01B can cause uveitis. (2) The manufacturing of the Shingrix vaccine, which uses Chinese hamster ovary (CHO) cells, may include trace levels of host cell proteins (HCPs) in the finished vaccine, which could cause autoimmune reactions [64]. In the existing case reports, due to limitations in detection methods, only some patients underwent genotyping, and no patients who received the subunit herpes zoster vaccine were found to have the Oka strain positive genotype. This suggests that the possible cause of ocular complications in patients after receiving the recombinant subunit vaccine may be immune modulation phenomena leading to the reactivation of dormant VZV [60].

The latest article proposes a new hypothesis suggesting that the injection site of vaccination may be related to ocular complications following vaccination [74]. There may be a potential correlation between the pathophysiological mechanisms of upper limb vaccination and post-vaccination ocular complications. Proximal viral infections—those closer to the head—can allow the virus to directly reach the retina through the central retinal artery, which is a branch of the ophthalmic artery that originates from the internal carotid artery. It suggests that retinitis requires proximal viremia after vaccination closer to the head, allowing the virus to reach the trigeminal ganglion or superior cervical ganglion. Vaccination in the thigh would reduce the likelihood of the virus reaching its target ganglia. In many countries, Varivax is given to young children by injection into the thigh rather than the arm. Young children who were given Varivax by injection into the leg did not have any ocular complications. In the currently reported cases, except for one 5-year-old child, all other cases were older than 9 years. For this 5-year-old child, the injection site was clearly documented as the right upper limb. There were no cases of ocular complications in children under the age of 9 years, even though many million doses of Varivax have been given to young children under the age of 9 years, often in the leg.

## 4. Conclusions

All things considered, more research is necessary to fully understand the connection between the VZV vaccine and ocular problems. Half of adults who reach the age of 85 will suffer from HZ, and one-third of adults will develop it throughout their lives. HZ has a significant negative influence on older people, despite the fact that it is rarely fatal. It causes persistent neuropathic pain, diminished physical and social functioning, emotional discomfort, decreased productivity, medical expenses, and irreversible loss of independence. The most economical methods of reducing disease incidence have been found to be preventative vaccines against concurrent conditions. Although ocular side effects after VZV vaccine are possible, they are quite uncommon, and immunization is advised.

## Data Availability

Data are available on request.

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
