# Peer review of "Varicella-Zoster Virus Infection and Varicella-Zoster Virus Vaccine-Related Ocular Complications"

_vaccines, 2025, doi:10.3390/vaccines13080782_

Round 1
Reviewer 1 Report
Comments and Suggestions for Authors
The authors in this article examine the potential relationship between the Varicella-zoster virus-associated vaccine and shingles and chickenpox, with herpes zoster presenting higher risks in elderly individuals. The review focuses on ocular complications and postherpetic neuralgia, with an emphasis on the safety and importance of vaccination.
The review is intriguing and potentially quite exciting. However, there are a few essential points that need to be addressed.
- [ Lines 16 to 20] Please provide recent references.
- The discussion and background sections lack sufficient information regarding the mechanisms of reactivation and latency of VZV, as well as the clinical implications of viral reactivation. Greater elaboration on how vaccination impacts VZV latency and whether it may be helpful in preventing or modulating reactivation would enhance the completeness and pertinence of the review.
Author Response
Varicella-Zoster Virus (VZV) is a neurotropic human herpesvirus and a double-stranded DNA virus that can cause two distinct diseases: varicella (chickenpox) and herpes zoster (shingles), a vesicular dermatomal rash brought on by the latent virus reactivating. Varicella zoster virus-specific T cell-mediated immunity is largely in charge of regulating viral reactivation and causing herpes zoster. Months to years after the initial infection resolves, VZV establishes latency in peripheral ganglia neurons. Endogenous (subclinical) reactivation and exogenous reexposure to VZV periodically increase immunity, and VZV-specific memory T cells, which have a mixed central and effector nature, are crucial for preserving VZV latency(1). A latent virus can nevertheless replicate and produce recurring clinical illness (2). Declining T cell immunity, not humoral immunity, is primarily responsible for the rise in VZV reactivation seen with aging (3). Regardless of varicella incidence, herpes zoster is a rare, year-round condition with no seasonal predominance. There is no proof that HZ is caused by an external VZV infection, meaning that it can be contracted by coming into contact with a HZ or varicella carrier. Instead, variables affecting the virus-host relationship—most notably the host's VZV-CMI, which preserves VZV latency—determine the prevalence of HZ. Even in nations where varicella vaccination has all but eradicated the disease, a number of epidemiologic studies have demonstrated that the age-specific incidence of HZ has risen during the previous seven decades(4-7).
HZ has a lifetime risk of more than 30%. The incidence rate of HZ in people aged 50 and older varies from 5.23 to 10.9/1000 person-years(8) and from 3 to 5/1000 person-years globally(9). As people age, the frequency and severity of HZ rise. Nearly half of older adults with HZ experience problems(9), and 50% of those who live to age 85 will acquire HZ(9-11). Herpes zoster and its aftereffects can cause irreparable loss of independence and significantly lower quality of life(12). Postherpetic neuralgia (PHN), the most common disabling consequence of HZ, is characterized by neuropathic pain and dysesthesia that persists for weeks, months, or even years after the rash has healed(13-16). The frequency of PHN rises significantly with age(13, 17). The development of vaccinations to protect older adults from HZ is prompted by the rising incidence of HZ and its crippling consequences as people age. The FDA has approved a number of vaccines to prevent VZV. A live attenuated virus called Varivax is used to prevent varicella in children(18). In 2006, the FDA authorized Zostavax, a live attenuated Oka strain virus, for the secondary prevention of zoster. However, because of the risk of zoster reactivation in immunocompromised persons and its gradual decline in immunogenicity, its usage has been restricted(19). The FDA authorized Shingrix, a recombinant subunit zoster vaccine, in 2017 for the prevention of zoster in immunocompetent people 50 years of age and older. Clinicians should be aware of the possibility of post-vaccination VZV infection or reactivation, even if it seems to be uncommon. A pertinent overview of VZV infection and ocular problems linked to VZV vaccination is hence the goal of this work.
Reviewer 2 Report
Comments and Suggestions for Authors
Very interesting literature review on VZV infections and VZV vaccines related ocular complications.
Ocular complications of VZV are nicely explained, as well as reported and supposed ocular complications of VZV vaccines in the literature review.
The different mechanisms, infectious, immunologic , or both, are clearly explained.
Just one thing, there is a spelling mistake in Shingrix, sometimes written Shigrix, to be corrected of course.
The fact that ocular inflammation also could be secondary to vaccine adjuvant is very interesting.
In conclusion, nice article easy to read with many possible explanations about infection and inflammation.
Author Response
Comments: there is a spelling mistake in Shingrix, sometimes written Shigrix, to be corrected of course.
Response: I am sorry, I have already made the modifications in the manuscript. Thank you very much.
Reviewer 3 Report
Comments and Suggestions for Authors
The ocular complications of vaccination with live varicella and zoster vaccines have been overlooked for decades. Therefore, this manuscript is a useful addition to the vaccine adverse events literature. The authors have performed a very thorough search of the medical literature and assembled a valuable table of published cases. However, they may have overlooked a few reports that are mentioned below.
1.Table. The authors missed a case report of acute retinal necrosis after Varivax. Please add to the Table a report by U. Hayat, Kansas Journal of Medicine 13:324-325, 2020. PMID: 33343828. The virus in this case was tested and found to be vaccine strain.
2.Table. The authors missed a second case report of uveitis after Varivax. This young child had been given Varivax in the arm. Please add to Table a report by C. Andrade et al, Frontiers in Pediatrics, 13:1567164, 2025. PMID: 40309166
3.Lines 103-106. Give 2 calculations for average age rather than one calculation. Give the average age of patients with complications after Varivax and the average age for patients with complications after Zostavax.
4.Conclusions, line 148. Perhaps because another review paper is so recent, the authors missed reading a recent review of the severe adverse events after Zostavax, including ocular complications. See review by P. Kennedy et al, Journal of Virology, volume 99, issue 2, February 2025, PMID: 39818965. The authors present a new hypothesis about ocular complications after Varivax and Zostavax. The ocular complications appeared only in patients who had been injected with Varivax or Zostavax in the upper arm. In many countries, Varivax is given to young children by injection into the thigh rather than the arm. Young children who were given varivax by injection into the leg did not have any ocular complications. In the current manuscript, there were no cases of ocular complications in children under the age of 9 years, even though many million doses of varivax have been given to young children under the age of 9 years, often in the leg. There is a new report of an ocular complication in a young child, but this child was vaccinated in the arm (Comment 2). Thus, this hypothesis about the arm and the leg may explain the difference in ages of people who develop ocular complications. Please discuss briefly in text.
Author Response
Comments1 and Comments2: Please add to the Table a report by U. Hayat, Kansas Journal of Medicine 13:324-325, 2020. PMID: 33343828. and add to Table a report by C. Andrade et al, Frontiers in Pediatrics, 13:1567164, 2025. PMID: 40309166
Response: I'm so sorry. I have added these two cases to the table. Thank you.
Comments3: Lines 103-106. Give 2 calculations for average age rather than one calculation. Give the average age of patients with complications after Varivax and the average age for patients with complications after Zostavax.
Response: I have separately calculated the average age of ocular complications following varicella vaccination and the average age following herpes zoster vaccination in the manuscript.
Comments4: This hypothesis about the arm and the leg may explain the difference in ages of people who develop ocular complications. Please discuss briefly in text.
Response: At the end of the manuscript, I reanalyzed and discussed the possible mechanisms of ocular complications after vaccination against varicella-zoster virus. Thank you.
u